# First Report of Fungal Pathogens Causing Leaf Spot on *Sorghum–Sudangrass* Hybrids and Their Interactions with Plants

**DOI:** 10.3390/plants12173091

**Published:** 2023-08-29

**Authors:** Junying Li, Jingxuan Xu, Hongji Wang, Changfeng Wu, Jiaqi Zheng, Chaowen Zhang, Yuzhu Han

**Affiliations:** 1College of Animal Science and Technology, Southwest University, Chongqing 402460, China; lijunying0312@163.com (J.L.); xujingxuan0128@163.com (J.X.); whongji123@163.com (H.W.); zjq2000@email.swu.edu.cn (J.Z.); 18651661447@163.com (C.Z.); 2Bazhong Academy of Agriculture and Forestry Sciences, Bazhong 636099, China; 15002391137@163.com; 3Institute of Biotechnology, Zhejiang University, 866 Yuhangtang Road, Hangzhou 310058, China; 4Chongqing Key Laboratory of Herbivore Science, Chongqing 402460, China

**Keywords:** *sorghum–sudangrass* hybrid, biological characteristics, *Nigrospora sphaerica*, *Colletotrichum boninense*, *Didymella corylicola*, metabolomics analysis

## Abstract

The *sorghum–sudangrass* hybrid is the main high-quality forage grass in Southwest China, but, in recent years, it has suffered from leaf spot disease, with a prevalence of 88% in Bazhong, Sichuan, China, seriously affecting yield and quality. The causal agents were obtained from symptomatic leaves by tissue isolation and verified by pathogenicity assays. A combination of morphological characterization and sequence analysis revealed that strains SCBZSL1, SCBZSX5, and SCBZSW6 were *Nigrospora sphaerica*, *Colletotrichum boninense*, and *Didymella corylicola*, respectively, and the latter two were the first instance to be reported on *sorghum–sudangrass* hybrids in the world. SCBZSX5 significantly affected the growth of the plants, which can reduce plant height by 25%. The biological characteristics of SCBZSX5 were found to be less sensitive to the change in light and pH, and its most suitable culture medium was Potato Dextrose Agar (PDA), with the optimal temperature of 25 °C and lethal temperature of 35 °C. To clarify the interactions between the pathogen SCBZSX5 and plants, metabolomics analyses revealed that 211 differential metabolites were mainly enriched in amino acid metabolism and flavonoid metabolism. *C. boninense* disrupted the osmotic balance of the plant by decreasing the content of acetyl proline and caffeic acid in the plant, resulting in disease occurrence, whereas the *sorghum–sudangrass* hybrids improved tolerance and antioxidant properties through the accumulation of tyrosine, tryptophan, glutamic acid, leucine, glycitein, naringenin, and apigetrin to resist the damage caused by *C. boninense*. This study revealed the mutualistic relationship between *sorghum–sudangrass* hybrids and *C. boninense*, which provided a reference for the control of the disease.

## 1. Introduction

The *sorghum–sudangrass* hybrid is an annual grass crop that is a cross between *Sorghum bicolor* and *Sorghum sudanense*, which is drought-resistant, cold-tolerant, barrenness-tolerant, stress-resistant, high-yield, high in sugar content, high-tillering, regenerative, and rich in foliage [1]. It is widely grown worldwide because it has abundant fodder values and can be harvested for green fodder, silage, or hay. However, domestic and international research on *sorghum–sudangrass* hybrids has mainly focused on crossbreeding and yield improvement [2,3,4,5], with fewer reports on associated diseases. The available studies include a case of Maize dwarf mosaic virus infecting *sorghum–sudangrass* hybrids in North America [6]. In addition, rust disease of *sorghum–sudangrass* hybrids in high humidity has been reported in China [7], where the disease is mainly characterized by orange–red leaves with round or oval spores. There has been no research on leaf spot disease regarding *sorghum–sudangrass* hybrids in China. This plant is susceptible to fungal infection with leaf spots in Bazhong City, Sichuan Province, China, which is located at an altitude of 1000–2000 m, with high temperatures and rainy summers. In 2022, we found that the incidence of *sorghum–sudangrass* hybrids leaf spot disease in Bazhong City was extremely high, with an average of 88%. As this forage is one of the most important pasture grasses in Sichuan Province, leaf spot disease induces a serious impact on the development of animal husbandry.

The typical symptoms on the leaves of *sorghum–sudangrass* hybrids included long oval spots with a white center and reddish-brown margins starting from the edge of the leaves, and the disease was more serious than on the rhizomes. Zhu et al. [8] discovered that *Nigrospora sphaerica* caused leaf spot disease on *sorghum–sudangrass* hybrids; there are almost no other related reports on this plant. Therefore, we obtained the relevant agents through tissue isolation and identified the pathogens morphologically and molecularly after pathogenicity assays, combined with the results of their pathogenic damage and biological characteristics. The most severe pathogenic damage and the most environmentally adapted pathogens were then selected for metabolomics analysis to reveal the interactions between the pathogens and the plants. This study provides theoretical evidence for the control of leaf spot disease of *sorghum–sudangrass* hybrids as well as offers a reference regarding the interactions between *sorghum–sudangrass* hybrids and the pathogenic fungi.

## 2. Results

### 2.1. Evaluation of the Pathogenicity of 3 Pathogenic Fungi

In this study, three fungi were isolated from the diseased *sorghum–sudangrass* hybrids, including SCBZSL1 (28.57%), SCBZSX5 (21.43%), and SCBZSW6 (21.43%). *Sorghum*–*sudangrass* hybrids (purchased from Jiangsu Zhengda Grass Industry, Liyang, China) were planted in the pasture base of Southwest University, and the pathogenicity was verified after 3 weeks of cultivation of *sorghum–sudangrass* hybrids. The leaves and rhizomes of *sorghum–sudangrass* hybrids inoculated with the test strains showed lesions after 3 days of inoculation, and the incidence rate was 100%. The 7-day-old *sorghum–sudangrass* hybrids were observed. It was found that the lesions at the stem and leaf inoculation sites of *sorghum–sudangrass* hybrids treated with SCBZSL1 and SCBZSX5 were the same as those of the naturally diseased plants. The symptoms of *sorghum–sudangrass* hybrids inoculated with SCBZSW6 were slightly different from those in natural onsets. The lesions were smaller and only included red lesions. The diseased plants were re-isolated, and the pathogenic fungi consistent with the morphological characteristics of the inoculated strains were obtained. When compared with healthy plants, the development degree of *sorghum–sudangrass* hybrids after inoculation with different pathogens was affected to varying degrees (Figure 1). The growth of *sorghum–sudangrass* hybrids inoculated with SCBZSX5 was the most significantly inhibited, and its average plant height was only 84.29 cm. Number of rhizomes, root bulk density, plant height, and root length were significantly lower than those of healthy plants (Table 1). The symptoms of SCBZSL1 and SCBZSX5 were similar. The growth indexes [9] (number of rhizomes, root bulk density, plant height, and root length) of SCBZSL1-susceptible plants were lower than control groups but significantly better than those inoculated with SCBZSX5. SCBZSW6-susceptible plants exhibited typical symptoms but had the least effect on the growth of *sorghum–sudangrass* hybrids.

### 2.2. Morphological Identification

The pathogenic fungi were inoculated on PDA plates and cultured in an incubator (Shanghai Chuchi Biotechnology Co., Shanghai, China) at 25 °C for 3 d and 7 d. The colony of SCBZSL1 was initially white and later turned black, with abundant aerial hyphae. The mycelium was long, smooth, and transparent, and the back of the matrix was slightly yellow. The meristematic cells were light brown and spherical, and the conidia were black, shiny, smooth, spherical, or oval, and had many branches (Figure 2c–e). The morphological characteristics of SCBZSL1 were consistent with the description of *Nigrospora sphaerica* [10]. After SCBZSX5 was cultured for 7 days, the colony was white, the central part was light brown with protrusions, and the back of the colony was light yellow. Conidia were pale gray, oblong, longitudinal, and transverse diaphragm, and mycelia were dendritic and branched (Figure 3c–e). It had the typical characteristics of *Colletotrichum boninense* [11]. After 7 days of culture, the colony of SCBZSW6 was brown, and the colony had concentric rings. The inner ring was slightly black–brown, the middle ring gradually changed from black- brown to yellow–green, the outer ring was gray, and the outset ring was light brown. The inner ring and the middle ring colony had folds. The back of the substrate was black–brown, the outer ring was gray, and there were cross cracks in the later stage of culture. Conidia were transparent, ellipsoid to oblong, with diaphragm. Mycelial was slightly protuberant (Figure 4g). The morphology of SCBZSW6 in this study agreed with previous descriptions of *Didymella corylicola* [12]. From the colony culture characteristics, spore morphology, hyphae characteristics, color, and size of the test strains, the three test strains were *N. sphaerica*, *C. boninense*, and *D. corylicola*.

### 2.3. Phylogenetic Tree Analysis

All the obtained sequences were input into GenBank BLAST for homologous sequence search to find strains with high similarity (Table 2). The ITS and BT sequences of the strain SCBZSL1 were deposited in GenBank, and the accession numbers OR243751 and OR250667 were obtained. The ITS, BT, and ACT gene sequences of the strain SCBZSX5 were submitted to GenBank with accession numbers OR243752, OR250666, and OR250668. The ITS, LSU, and BT gene sequences of the strain SCBZSW6 were submitted to GenBank with accession numbers OR243778, OR248151, and OR250665. Construct a phylogenetic tree using the ITS and BT sequences of SCBSZL1, the ITS, BT, and ACT sequences of SCBZSX5, and the ITS, LSU, and BT sequences of SCBZSW6, respectively, with the relevant reference sequences in Table 2. *N. sphaerica*, *C. boninense*, and *D. corylicola* isolates formed clades that clustered together with SCBZSL1, SCBZSX5, and SCBZSW6, respectively (Figure 5, Figure 6 and Figure 7). These clades were supported by 100% bootstrap values, which confirmed the identification of three pathogenic strains.

### 2.4. Effect of Different Light on the Growth of the Test Strains

All strains had strong adaptability to light duration, and the morphology of pathogens changed as illuminated by different light regimes (Figure 8). The 0 hD/24 hL condition was the most suitable for the growth of the three strains. Under this condition, the morphology of SCBZSW6 changed, the overall colony became gray–white, the middle was yellow with black spots, and the back matrix did not change. The color of the SCBZSX5 colony changed from light cream to darker yellow. When cultured at the light condition of 24 hD/0 hL, the growth speed of SCBZSL1 significantly decreased, and the mycelium of the middle layer became thinner. A semi-transparent ring appeared around the colony of SCBZSX5. Under the condition of 12 hD/12 hL, the mycelium of SCBZSL1 was the densest, and the back matrix was annually ring-shaped, and no cross crack was found. SCBZSL1 and SCBZSW6 did not produce spores under three light conditions (Table 2). The effects of the illumination duration on SCBZSX5 were significantly different (*p* < 0.05).

### 2.5. Effect of Different Temperatures on the Growth of the Test Strains

The colony diameter of the tested strains SCBZSL1, SCBZSX5, and SCBZSW6 increased first and then decreased in the temperature range of 5–35 °C (Figure 2l, Figure 3l and Figure 4l), and the colony morphology did not change. When the temperature reached 5 °C or 35 °C, SCBZSX5 and SCBZSW6 did not grow. It can be seen that 20–25 °C was the suitable temperature range for the growth of SCBZSL1 and SCBZSW6, the optimum temperature of these two strains was 25 °C, and the average colony diameter after inoculation of 7 days was 8.1 cm and 6.4 cm, respectively. The suitable temperature range for the growth of SCBZSX5 was 20–30 °C, the optimum growth temperature was 25 °C, and the average colony diameter was 8.7 cm on the 7th day. In the suitable temperature range for the growth of three pathogens, the average colony diameter of the strains gradually rose with the increase in temperature. SCBZSL1 and SCBZSW6 did not produce spores after being cultured at 5–35 °C for 7 days. SCBZSX5 produced spores only at 5 °C, 15 °C, 25 °C, and 30 °C (Table 2). SCBZSX5 produced the most spores at 25 °C, but there was no significant difference compared with the number of spores at 30 °C (*p* > 0.05). It is worth noting that low temperature (5 °C) could delay the growth rate of SCBZSX5, while the spore production was inhibited at the temperature of 10 °C. This situation may suggest the climatic characteristics of the peak incidence of *sorghum–sudangrass* hybrids.

### 2.6. Effect of Different Media on the Growth of the Test Strains

The growth status of the three strains on the tested medium was significantly different (Figure 9). SCBZSL1, SCBZSX5, and SCBZSW6 obtained the fastest growth rate on PDA plates, and the average diameters of 7-day-old colonies were 8.1 cm, 8.7 cm, and 6.4 cm, respectively. When cultured on Salt Czapek Dox Agar (SCDA), the growth speed of the three pathogens was the slowest, and the diameters of the 7-day-old colonies were 1.1 cm, 2.6 cm, and 0.8 cm, respectively. This indicates that the three pathogens have higher utilization efficiency of glucose. Compared with the morphology of SCBZSL1 cultured on PDA, the mycelium of SCBZSL1 cultured on Oatmeal Agar (OA) and Potato Saccharose Agar (PSA) was denser and the middle part was villous. The mycelial of SCBZSX5 changed significantly on different media. The colonies cultured by PSA were white, very short villi, and wrinkled, while the morphology on Malt Extract Agar (MEA) and OA culture showed snowflake folds, short villi, and denser hyphae accompanied by small voids. As for SCBZSW6, the colonies on PSA, MEA, and OA exhibited different characteristics. The colonies of SCBZSW6 cultured in PSA were brown and bark-like folds, the ones cultured in MEA were yellow–green in the middle and gray–white in the periphery, while the colonies cultured in OA were gray–white with dark gray marks on the surface. SCBZSL1 and SCBZSW6 also did not produce spores. SCBZSX5 produced spores only when cultured in PDA and PSA, and the number of spores produced by SCBZSX5 cultured in PDA was the highest (6.8 × 10^4^ cfu/mL).

### 2.7. Effect of pH on the Growth of the Test Strains

Three pathogens could grow within the tested pH range, indicating that they had strong adaptability, and the growth speed was greatly affected by the pH values. As the pH value increased, the growth speed of SCBZSL1 improved first and then decreased. When the pH reached 7, the growth rate of SCBZSL1 was the fastest, and it also suggests that SCBZSL1 was neither acid-resistant nor alkali-resistant. SCBZSX5 grew well in the tested pH range and was affected little by pH changes, implying that this strain was resistant to both acid and alkali. The colony diameter of SCBZSW6 decreased when the pH value was increased, indicating that the tested strain SCBZSW6 was resistant to acid and alkali. SCBZSL1 and SCBZSW6 did not produce spores under all pH conditions. SCBZSX5 produced spores at pH 5, 7, and 9 (Table 3), and the sporulation decreased with the increase in pH, which was in line with the changing trend in colony diameter. When the pH value was 5, the number of spores produced by SCBZSX5 was the most (1.4 × 10^5^ spores/mL).

### 2.8. Effect of Pathogenic Fungi on Plant Metabolism

To explore the interaction between SCBZSX5 and *sorghum–sudangrass* hybrids, we analyzed the whole metabolome between infected *sorghum–sudangrass* hybrids and healthy plants [13]. The PCA score map (Figure 10a) showed that the PC1 and PC2 accounted for 58.42% and 30.24% of the total variance, respectively, indicating that there were significant differences in the metabolic profiles of different groups of samples. A total of 777 different metabolites were identified by non-targeted metabolomics analysis, and 211 differential metabolites were screened with *p* value < 0.05 and VIP > 1. These differential metabolic species were divided into seven categories, containing lipid and lipid molecules, phenylpropanoids and polyketides, organic heterocyclic compounds, organic acids and their derivatives, benzene and organic oxygenates, and other substances (Figure 10c). Compared with the control group (CK), 148 metabolites were up-regulated, and 63 metabolites were down-regulated in the diseased group (TR) (Figure 10b). MetaboAnalyst 5.0 was used to create biological metabolic pathways, and an effect value > 0.1 was considered to be a potential target pathway. The KEGG database was applied to analyze the main enrichment metabolic pathways of differential metabolites, including arginine biosynthesis, D-glutamine and D-glutamate metabolism, nitrogen metabolism, phenylalanine metabolism, phenylalanine, tyrosine, and tryptophan biosynthesis (Figure 10d). The heat map directly demonstrated the changes in each differential metabolite. Organic acids, which are the key metabolite in the metabolic pathway and are primarily involved in the plant’s glucose metabolism and other cycles, are more prevalent in the enriched species of KEGG. Lipids and lipid molecules are also the predominated metabolites of the plant, mainly for energy supply and transformation into a variety of physiologically active molecules involved in the response of organisms, and palmitoleic acid is a typical example. In addition, the biosynthesis of phenylpropanoids and polyketides also accounts for a large part. Flavonoids like hesperetin and daidzein are secondary metabolites that are widely present in plants and play a defensive role against biological stresses. The diagram (Figure 10e–g) shows three types of metabolites with significant alterations in phenylpropanoids and polyketides, lipids and lipid molecules, organic acids, and their derivatives. Compared with the CK group, 57 metabolites were significantly up-regulated in the diseased plants. Among them, the amount of oleic acid glycidyl ester, formononetin, daidzein, daidzein, and β-leucine varied most greatly and was found to be more than twice those in the CK group. Eighteen metabolites were significantly down-regulated, and Cianidanol was down-regulated up to 0.02 times. In addition to these three kinds of substances, other compounds with important functions have also changed significantly in the diseased *sorghum–sudangrass* hybrids. For example, quinic acid, chlorogenic acid, Hex-2-ulose, glutara, and Ketoconazole l were all down-regulated.

A stress-related metabolic pathway map was constructed to fully understand the changes in metabolites in *sorghum–sudangrass* hybrids under pathogen infection (Figure 11). The dominant pathways include amino acid metabolism, flavonoid synthesis, and tricarboxylic acid cycle. Seven metabolites related to amino acid metabolism were identified, consisting of serine, tryptophan, leucine, tyrosine, glutamic acid, acetyl proline, and leu-arg. Compared with the control group (CK), tryptophan was obviously up-regulated and pyruvate was significantly accumulated. Pyruvate is converted into acetyl-CoA into the tricarboxylic acid cycle. The metabolites involved in the TCA cycle are citric acid, α-ketoglutarate, succinic acid, fumaric acid, and oxaloacetate. Among them, glutamic acid up-regulation allows for substantial accumulation of α-ketoglutarate. In the flavonoid synthesis pathway, coumarin and naringenin were considerably up-regulated, and related flavonoids were highly amassed to respond to pathogen stress.

## 3. Discussion

As an important high-quality forage grass in southwest China, the disease control of *sorghum–sudangrass* hybrids is very important to improve economic benefits. In this study, three fungi were isolated from *sorghum–sudangrass* hybrids, including *N. sphaerica*, *C. boninense*, and *D. corylicola*. The pathogenicity assays demonstrated that all three fungi infected *sorghum–sudangrass* hybrids. Among them, SCBZSW5 (*C. boninense*) induced the most serious damage to *sorghum–sudangrass* hybrids, and number of rhizomes, root bulk density, plant height, and root length were greatly affected. In previous reports, *N. sphaerica* has been shown to inhibit the growth of *sorghum–sudangrass* hybrids [8]. The sexual type of *C. boninense* belongs to the phyla Ascomycota, subphyla pezizomycotina, class sordariomycetes, subclass sordariomycetidae, and glomerellaceae [14]. *C. boninense* widely causes serious diseases in many plants. For example, it induces brown–black and red peel spots and soft rot of avocado in the state of Michoacán, Mexico, which severely reduced avocado productivity [15]. In Colombia, it widely invaded plants, especially the tree tomatoes and mangoes [16]. Tree tomatoes are mainly produced in Cundinamarca, Antioquia, Caldas, and Nariño, with a total planting area of about 4500 hectares. As much as 50% of the production of tree tomatoes in these states is cut by the pathogen. Mango is widely cultivated in Tolima State, where the loss of mango production is as high as 60% [16]. According to Marzia Scarpari et al. [12], *D. corylicola* represents a fungal species associated with hazelnut fructification, which has extensive adaptability to temperatures and can even grow at 5 °C. To our knowledge, this is the first report of the disease caused by *D. corylicola* on *sorghum–sudangrass* hybrids.

In this study, *C. boninense* was the main pathogen of *sorghum–sudangrass* hybrids leaf spot. The growth of the mycelium was not affected by the pH value. The optimum temperature for colony growth was 25 °C. The colony growth rate was positively correlated with temperature before 25 °C, and negatively correlated above 25 °C, which was consistent with the findings of Chung et al. [17]. There was no significant difference in growth rate under three different culture conditions of 24 hD/0 hL, 0 hD/24 hL, and 12 hD/12 hL, which was in accordance with the results of *C. gloeosporioides* reported by Zhao Jie et al. [18].

This study explored the sporulation of *C. boninense* under different conditions. The results showed that the optimum conditions for sporulation of *C. boninense* were at the temperature of 25 °C with a pH of 7 and PDA as a nutrient substrate in full darkness. Chen et al. [19] reported that the optimum temperature for sporulation of *C. gloeosporioides* was 28 °C, which was similar to the results of this study. In the study of Chen et al., light could promote sporulation, which was contrary to the results of our experiment. The *C. gloeosporioides* come from the leaves of Debuti in Pingxiang, Guangxi. Zhao et al. [20] reported that the suitable light condition for sporulation of the same genus of *C. gloeosporioides* was dark, which was consistent with the results of this study. The *C. gloeosporioides* originate from Houttuynia cordata in Ya’an, Sichuan Province; this indicates that the conditions suitable for sporulation of *Colletotrichum* spp. May vary greatly with geographical environment and host. The biological characteristics of mycelial growth and sporulation of the fungus can provide an important reference for the subsequent study of the pathogenic mechanism and establish a basis for effective prevention and control. Conidia are the asexual reproduction mode of most Ascomycota and all Deuteromycetes, and their sporulation quantity is closely related to mycelial growth speed. The spores of *N. sphaerica* and *D. corylicola* were not observed under normal culture conditions. Spore production may be associated with the disease severity of *sorghum–sudangrass* hybrids because these fungi infected *sorghum–sudangrass* hybrids with similar lesions.

In order to further understand the interaction between plants and pathogens, metabolomics analysis showed that amino acid metabolic pathways were significantly enriched in diseased *sorghum–sudangrass* hybrids, including arginine biosynthesis, D-glutamine and D-glutamic acid metabolism, nitrogen metabolism, phenylalanine metabolism, phenylalanine, tyrosine, and tryptophan biosynthesis. Amino acids play an important role in the growth of plants [21]. In this study, the expression of acetyl proline and leucine arginine was significantly down-regulated, and the expression of amino acids such as tyrosine, tryptophan, glutamic acid, and leucine was evidently up-regulated. Previous studies have shown that high proline content can improve the salt tolerance of plants [22], and the down-regulation of acetyl proline expression in the metabolites of diseased plants may lead to salt-stress-induced lipid peroxidation in the leaves of *sorghum–sudangrass* hybrids. Therefore, the pathogen was expected to destroy the salt tolerance of plants by blocking the expression of acetyl proline and inducing plant disease. Researchers have found that the RxLR (Arg-any amino acid-Leu-Arg) effector PcAvh103 related to leucine arginine is related to the inhibition of plant immunity [23]. It is speculated that plants mitigate their suppression of plant immunity by down-regulating the expression of leucine arginine, thereby resisting the stress of pathogens. The elevation of tyrosine expression may be due to protein tyrosine phosphorylation, which is involved in the response of plants to stress signals [24]. Tryptophan is related to plant growth. In detail, relevant studies have shown that plants exposed to high salinity or drought exhibit a decrease in the expression of tryptophan synthase β subunit 1 (TSB1) [25], which reduces the accumulation of tryptophan and auxin, thereby inhibiting growth. Therefore, it is hypothesized that the up-regulation of tryptophan expression may be connected to the down-regulation of proline, leading to a reduction in plant salt tolerance and salt stress. In addition, glutamic acid and leucine are crucial for enhancing disease resistance or stimulating autoimmunity responses in plants. Kim et al. have revealed that glutamic acid has the function of reshaping plant microbial communities and enriching functional core microorganisms [26]. Plants treated with glutamic acid have a significant reduction in diseases caused by Botrytis and Fusarium [26]. The domain of leucine is the innate immunity of plants [27]. Flavonoid synthesis depends heavily on basic leucine zipper domain (bZIP) transcription factors, which are also vital for plants to adapt to varied stressful environments [27]. Tyrosine and tryptophan are also important precursors for the synthesis of secondary flavonoids and phytohormones [28].

As an essential secondary metabolite, flavonoids regulate a variety of processes in plant growth and development, and have antioxidant and anti-inflammatory properties [29]. Metabolomics analysis revealed 26 differential metabolites of flavonoids, among which 16 were significantly up-regulated and 10 were significantly down-regulated in diseased *sorghum–sudangrass* hybrids when compared with healthy plants. Glycitein possesses antioxidant and anti-allergic characteristics [30]. According to the study of Uchida et al. [31], inoculation of fungi will up-regulate Glycitein. In this study, it is speculated that *sorghum–sudangrass* hybrids can improve their antioxidant capacity by up-regulating Glycitein. Naringenin is a flavanone in flavonoids, which has been proven to have different effects on plant growth and metabolism. It enhances stress tolerance by regulating cell redox and ROS scavenging ability [32]. Apigetrin exhibits strong antifungal efficacy [33], and can considerably lower the potential of *Candida albicans* to establish biofilms. Based on the up-regulation of these substances, we can infer that plants have developed defense and resistance to pathogen infection. It is worth noting that there are some down-regulated differential metabolites in flavonoids, such as umbelliferone and caffeic acid. The C7 hydroxyl group of umbelliferone is beneficial to enhance the virulence of pathogenic fungi invading plants, and its toxicity can be greatly diminished by modifying the above sites [34]. The down-regulated umbelliferone suggested that the plant inhibits the virulence of pathogenic fungi by weakening the secretion of umbelliferone. Studies have shown that caffeic acid can improve the growth and yield of plants under drought stress [35]. In our study, regarding the reduced caffeic acid expression, it can be inferred that the role of pathogens destroyed this improvement pathway.

Lipids are essential components for plants [36,37,38]. In this study, it was found that the expression of palmitoleic acid was down-regulated. A recent study has shown that palmitoleic acid could induce fungi to create secondary spores, and these spores have the ability to infect the roots of the host plant and produce sporozoites [39]. Our results indicated that plants may inhibit the production of fungal spores by suppressing the expression of palmitoleic acid for defense. Jasmonic acid also has a defensive effect [40], and the defense response of maize seedlings can be improved by increasing jasmonic acid [41]. The expression of jasmonic acid was elevated, implying that jasmonic acid also has the same defense response in *sorghum–sudangrass* hybrids as maize seedlings. Spraying jasmonic acid is regarded as an effective approach to lower oleic acid content [42]. The down-regulation of palmitoleic acid expression and the up-regulation of jasmonic acid expression in our study may also be related to this reason.

## 4. Materials and Methods

### 4.1. Sample Collection and Fungal Isolation

In October 2022, leaf spot disease was observed on *sorghum–sudangrass* hybrids and sampled in Sichuan Province, China (97°21′–108°42′ E, 26°03′–34°19′ N), with a total of 56 locations in two cities (Chengdu and Bazhong) and three districts (Tongjiang, Bazhou, and Jinjiang). Using the chessboard sampling method, 50 samples were collected from each location, and 1050, 900, and 850 samples were obtained from each of the three regions, including 914, 684, and 748 diseased samples. The average incidence rates in Tongjiang, Bazhou, and Jinjiang districts were 87%, 76%, and 88%, respectively. Then, 280 disease samples were randomly selected from 56 regions, with 5 samples from each region, for pathogen isolation experiments. Infected tissues (3 mm × 3 mm) at the junction of disease and health of the leaf blades were cut, sterilized in 75% ethanol for 30 s, rinsed in sterile water, and then immersed with 1% sodium hypochlorite for 3 min, followed by rinsing and swabbing. The segments were then placed on PDA at 25 °C. After 3 d of incubation, mycelium growing at the edge was transferred to fresh PDA and cultured at 25 °C. All purified isolates were stored on PDA slant at 4 °C.

### 4.2. Pathogenicity Test

*Sorghum–sudangrass* hybrids were used to confirm the pathogenicity of three strains. *Sorghum–sudangrass* hybrids (purchased from Jiangsu Zhengda Grass Industry, Liyang, China) were planted in the pasture base of Southwest University. Use a punch with an inner diameter of 5 mm to drill holes at the edge of the colony after 7 days of PDA cultivation to obtain fungal cake. Inoculated on healthy leaves of *sorghum–sudangrass* hybrids that grow in the field for 3 weeks and are in the jointing stage. *Sorghum–sudangrass* hybrids have been previously disinfected with 75% ethanol and pricked with a sterile needle. The pathogens were applied to the wounded parts of the leaves; six replications were carried out for each strain. Healthy leaves and rhizomes that were punctured but not inoculated with the pathogen were taken as a blank control. The disease onsets were observed and recorded. Re-isolated pathogens were identified through colony and conidia morphology.

### 4.3. Morphological Identification

The pathogenic fungi were inoculated on PDA plates and cultured in an incubator (Shanghai Chuchi Biotechnology Co., Shanghai, China) at 25 °C for 3 d and 7 d. The morphology of the colonies was recorded. A small amount of mycelium was placed on a glass slide with a drop of water, and the characteristics of mycelium, conidia, and chlamydospores were observed under a microscope (Dongguan OSP Technology Co., Dongguan, China).

### 4.4. Molecular Identification and Phylogentic Analysis

Genomic DNA of the isolates was extracted according to the instructions of Fungal Genomic DNA Extraction Kit (Sobolite Technologies Ltd., Beijing, China). PCR amplification of ITS, TUB, ACT, and LSU regions was performed using primer pairs ITS1/ITS4, Bt-2a/Bt-2b, ACT512F/ACT783R, and LR0R/LR7, respectively (Table 4). All PCR amplifications were conducted in 25 μL reaction volumes. The system includes a template of 1 μL, upstream primers of 0.5 μL and downstream primers of 0.5 μL each, 2 × PCR Taq Master Mix of 12.5 μL, and ultrapure water of 10.5 μL. The PCR program was set as follows: pre-denaturation at 94 °C for 5 min, denaturation at 94 °C for 30 s, annealing at 54 °C for 30 s, and extension at 72 °C for 90 s, for a total of 35 cycles, and finally extension at 72 °C for 10 min, with a heat preservation of 4 °C. The amplification products were sent to GENEWIZ (Suzhou, China) for sequencing. DNA sequences obtained in this study were compared with corresponding sequences by BLAST searches in the National Center for Biotechnology Information (NCBI) database. The representative sequences were submitted in GenBank. MEGA 11 software and the maximum composite likelihood method were applied to build a phylogenetic tree.

### 4.5. Effect of Photoperiod on the Growth of the Test Strains

In order to understand the effect of photoperiod on the growth of pathogenic fungi, three different light conditions were set up, including 12 h of light and 12 h of darkness (12 hD/12 hL), 24 h light (24 hD/0 hL), and 24 h darkness (0 hD/24 hL). The pathogenic fungi were inoculated with PDA and cultured at 27 °C, and the colony diameter was measured after 3 d and 7 d in two perpendicular directions, and the spore production was determined by the hematological counting plate method [46]. Six replications were performed for each treatment.

### 4.6. Effect of Temperature on the Growth of the Test Strains

Plant diseases caused by fungi have certain climatic characteristics, and suitable temperatures can promote fungal infection of plants. In this study, seven different temperature gradients were set at 5 °C unit intervals, taking the annual temperature of Southwest China as a reference under the condition of a photoperiod of 24 h darkness (0 hD/24 hL). The test strains were inoculated on PDA plates and cultured at different temperatures (5 °C, 10 °C, 15 °C, 20 °C, 25 °C, 30 °C, and 35 °C) for 3 d. After 7 d, the colony diameter and spore production were measured at different temperatures, with six replications for each treatment.

### 4.7. Effect of Different Media on the Growth of the Test Strains

Fungi utilize carbon and nitrogen sources diversely. Thus, various culture media can directly affect the growth and spore production of fungi. Under the condition of a photoperiod of 24 h darkness (0 hD/24 hL), the test strains were placed in five different kinds of media, including PDA, PSA, SCDA, MEA, and OA, and cultured at 27 °C in an incubator. After inoculation of 3 d and 7 d, the size of colonies and quantity of spores were determined. Six replications were carried out for each treatment.

### 4.8. Effect of pH on the Growth of the Test Strains

pH is important for the growth and development of plants and pathogens. In this experiment, PDA was used as the basal medium, and four different gradients of pH were adjusted at 5.0, 7.0, 9.0, and 11.0 by adding 0.1 mol/L NaOH or HCl. The test strains were inoculated into the medium with different pH values and incubated at a photoperiod of 24 h darkness (0 hD/24 hL) and a temperature of 27 °C. The colony morphology and spore production were assessed after 3 d and 7 d. Each treatment was repeated six times.

### 4.9. Induced Spore Production Test

The strains SCBZSL1 and SCBZSW6 did not produce spores under the aforementioned culture conditions. In order to observe their spore morphology as well as mycelial characteristics, synthetic low-nutrient agar (SNA) was used to induce spore production. The ingredients of SNA include KH_2_PO_4_ (1.0 g/L), KNO_3_ (1.0 g/L), MgSO_4_·7H_2_O (0.5 g/L), KCl (0.5 g/L), glucose (0.2 g/L), sucrose (0.2 g/L), agar (15.0 g/L), pH (7.0).

### 4.10. Metabolomics Analysis

Samples from both groups of diseased plants (TR) and healthy plants (CK) were freeze-dried 50 g each in a freeze dryer (Beijing Songyuan Huaxing Co., Beijing, China) for 24 h and grounded with a high-throughput tissue grinder (Ningbo Xinzhi Biotechnology Co., Ningbo, China) at 50 Hz. Then, take 50 mg powder samples from each group and added to 0.6 mL of 70% methanol and allowed to stand at 4 °C for 12 h. After ultrasonic crushing for 5 min (25–40 KHz), the sample was centrifuged at 12,000× *g* for 10 min. Three biological replicates were set up for each group. Quality control (QC) samples were prepared by mixing 10 μL of TR samples and CK samples. One QC sample was inserted in every 3 samples to monitor method stability and data reliability.

LC-MS/MS analysis was performed using ultra-high performance liquid chromatography (UHPLC) UltiMate 3000 (Dionex, Sunnyvale, CA, USA) fitted with a UPLC Hypersil GOLD C18 column (2.1 × 100 mm, 1.9 μM particle size) (Thermo Fisher Scientific, Waltham, MA, USA) and coupled to a Q-Exactive Orbitrap (Thermo Fisher Scientific). The analytical conditions were as follows: flow rate, 0.2 mL/min. Column temperature, 35 °C. Injection volume, 2 μL. Mobile phase contained solvent A (ultrapure water containing 0.1% formic acid) and solvent B (0.1% formic acid methanol solution), solvent C (ultrapure water containing 0.1% NH_3_), and solvent D (0.1% NH_3_ in methanol). Gradient program (positive ion mode), 5% B and 95% A at 0–10 min, 5% A and 95% B at 10–12 min, 5% A and 95% B at 12–13 min, and 95% A and 5% B at 13.1–14 min. Gradient program (negative ion mode), 95% C and 5% D at 0–2.5 min, 95% D and 5% C at 2.5–16.5 min, 5% C and 19% D at 16.5–19 min, 95% C and 5% D at 19–20 min.

The Exactive Orbitrap was used to acquire MS/MS spectra in Information Dependent Acquisition (IDA) mode under the control of the acquisition software Xcalibur 7.9.11.1 (Thermo Fisher Scientific, Waltham, MA, USA). The HESI source was operated with the following parameters: sheath and auxiliary gas flow rates of 40 and 10 Arb, respectively [47], and a capillary temperature of 320 °C. The HESI source was operated with the following parameters: full mass spectral scan (*m*/*z* 70–1050) with a resolution of 70,000, MS/MS scanning mode was set to data-dependent ms2 (dd-ms2) scanning with a resolution of 35,000, and high collisional dissociation was set to 20/40/60 eV in NCE mode [47]. The spray voltage was 3.5 kV (positive ion mode)/–2.5 kV (negative ion mode).

### 4.11. Statistical Analysis

The data obtained from experiments on different physiological characteristics of plants and pathogens were analyzed by one-way ANOVA using Excel 2019 and GraphPad Prism 8.0. Metabolomic data were imported into Compound Discoverer 3.2 and after peak detection, extraction, deconvolution, normalization, and peak alignment were matched through Mz Cloud and mzVault databases. Using the online graphing tool Metware Cloud (https://cloud.metware.cn, accessed on 12 May 2023) and GraphPad Prism 8.0 software, potential differential metabolites were screened according to different projected significance (VIP > 1 and *p* < 0.05), and volcano plots and heat maps were generated. Differential metabolism chemicals were analyzed using Metaboanalyst 5.0 (https://www.metaboanalyst.ca/, accessed on 15 May 2023) for important metabolic pathways. In addition, the KEGG database (https://www.kegg.jp/kegg/pathway.html, accessed on 25 May 2023) was used to annotate and construct the pathway).

## 5. Conclusions

*N. sphaerica*, *C. boninense*, and *D. corylicola* were identified as the pathogens of leaf spot disease by morphological observation and molecular analysis. In particular, this is the first report of *C. boninense* and *D. corylicola* on *sorghum–sudangrass* hybrids in the world. Based on the determination of physiological indexes, it was found that *C. boninense* was the most serious causal agent of leaf spots on *sorghum–sudangrass* hybrids. After inoculating with *C. boninense*, the plant height, root development degree, and root volume dramatically decreased. The study of biological characteristics showed that *C. boninense* had the strongest environmental adaptability among the three strains. In order to further reveal the interaction between *C. boninense* and *sorghum–sudangrass* hybrids, metabolomics analysis indicated that *C. boninense* could destroy the salt tolerance of plants, which was manifested in the significant decrease in acetylproline and caffeic acid in infected plants. *Sorghum–sudangrass* hybrids can resist pathogen invasion by enhancing amino acid metabolism and flavonoid accumulation; for example, the amount of tyrosine and umbelliferone increased. The results of identification, biological characteristics, and metabolomics analysis of the pathogens on *sorghum–sudangrass* hybrids provide valuable information for crop yield and environmental safety. Furthermore, the interaction mechanism between *C. boninense* and *sorghum–sudangrass* hybrids described in this paper will provide novel perspectives for the prevention and control of leaf spot disease in *sorghum–sudangrass* hybrids.

## Figures and Tables

**Figure 1 plants-12-03091-f001:**
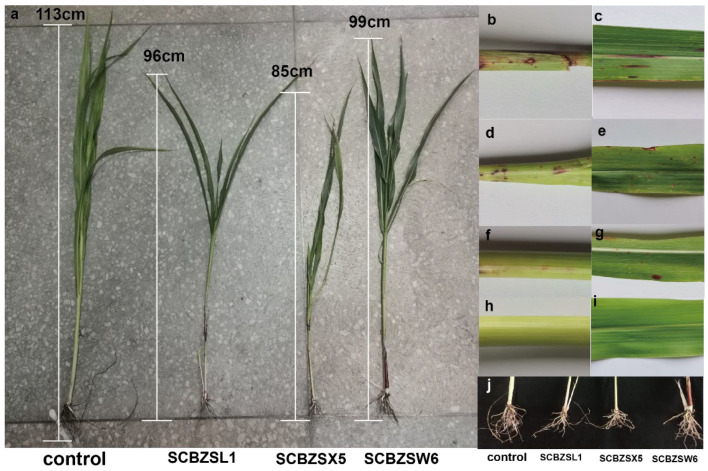
(**a**) Plant height of healthy *sorghum–sudangrass* hybrid and diseased *sorghum–sudangrass* hybrid inoculated with three pathogens. (**b**,**c**) SCBZSL1 was inoculated back to the stems and leaves of the diseased *sorghum–sudangrass* hybrids. (**d**,**e**) SCBZSX5 was inoculated back to the diseased *sorghum–sudangrass* hybrids’ stems and leaves. (**f**,**g**) SCBZW6 was inoculated back to the diseased stems and leaves of *sorghum–sudangrass* hybrids. (**h**,**i**) Healthy *sorghum–sudangrass* hybrids’ stem and leaf. (**j**) The root development of healthy *sorghum–sudangrass* hybrids and plants inoculated with three pathogens.

**Figure 2 plants-12-03091-f002:**
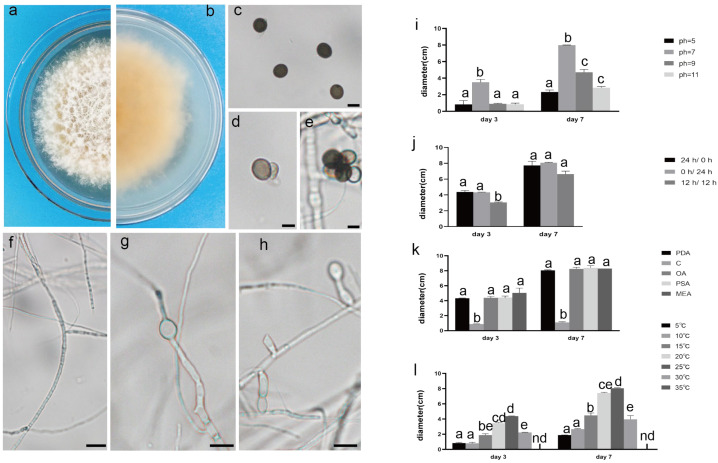
The front and back morphology of *N. sphaerica*. (**a**,**b**) PDA cultured for 7 days. (**c**) Conidia. (**d**,**e**) Meristematic cells and spores. (**f**–**h**) Different forms of mycelium, mycelium branching. (**i**) The average colony diameter of *N. sphaerica* cultured for 3 d/7 d under different pH conditions. (**j**) The average colony diameter of *N. sphaerica* cultured for 3 d/7 d under different light duration. (**k**) The average colony diameter of *N. sphaerica* cultured in different media for 3 d/7 d. (**l**) The average colony diameter of *N. sphaerica* cultured for 3 d/7 d under different temperature conditions. Gauge = 10 μM. The same lowercase letter in the chart indicates that the difference is not significant (*p* > 0.05), the difference is significant (*p* < 0.05), and the nd indicates that it is below the minimum scale or resolution.

**Figure 3 plants-12-03091-f003:**
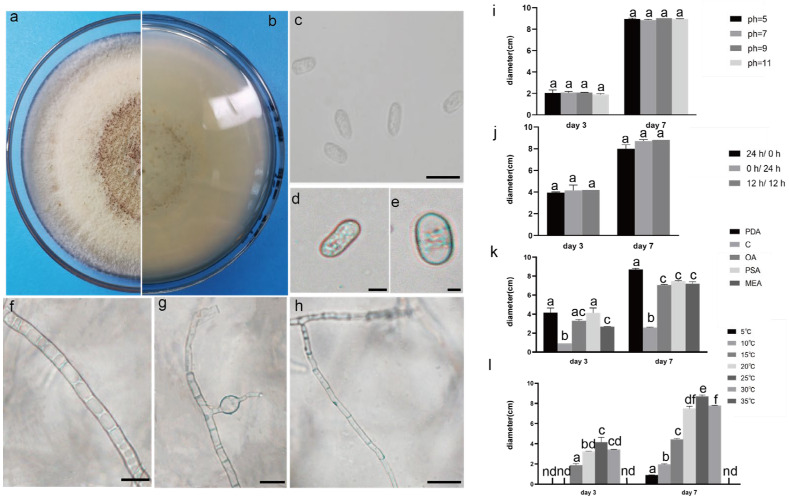
The front and back morphology of *C. boninense.* (**a**,**b**) PDA cultured for 7 days. (**c**–**e**) Conidium. (**f**) Smooth mycelium. (**g**) Hyphal protrusions. (**h**) Mycelium branched into acute angle. (**i**) The average colony diameter of *C. boninense* cultured for 3 d/7 d under different pH conditions. (**j**) The average colony diameter of *C. boninense* cultured for 3 d/7 d under different light durations. (**k**) The average colony diameter of *C. boninense* cultured in different media for 3 d/7 d. (**l**) The average colony diameter of *C. boninense* cultured for 3 d/7 d under different temperature conditions. Gauge = 10 μM. The same lowercase letter in the chart indicates that the difference is not significant (*p* > 0.05), the difference is significant (*p* < 0.05), and the nd indicates that it is below the minimum scale or resolution.

**Figure 4 plants-12-03091-f004:**
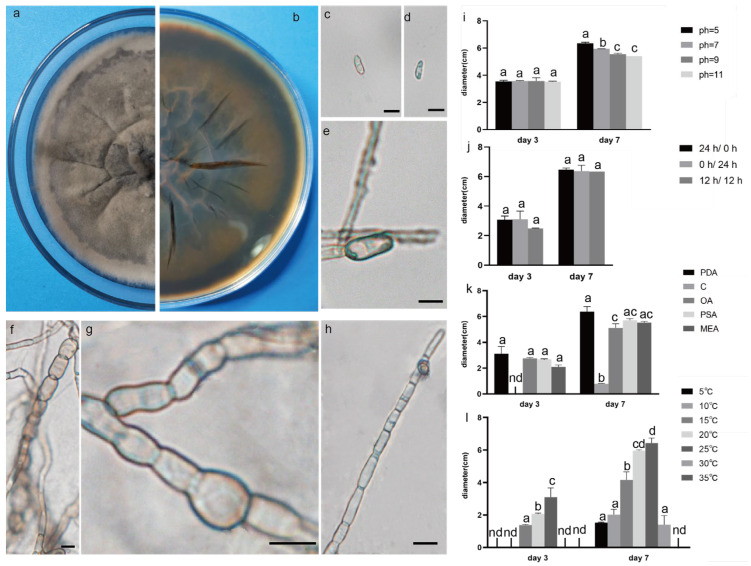
The front and back morphology of *D. corylicola.* (**a**,**b**) PDA cultured for 7 days. (**c**,**d**) Conidium. (**e**) Spores are forming. (**f**) Mycelium is chain. (**g**) Hyphal protrusions. (**h**) Spores are forming on hyphae. (**i**) The average colony diameter of *D. corylicola* cultured for 3 d/7 d under different pH conditions. (**j**) The average colony diameter of *D. corylicola* cultured for 3 d/7 d under different light durations. (**k**) The average colony diameter of *D. corylicola* cultured in different media for 3 d/7 d. (**l**) The average colony diameter of *D. corylicola* cultured for 3 d/7 d under different temperature conditions. Gauge = 10 μM. The same lowercase letter in the chart indicates that the difference is not significant (*p* > 0.05), the difference is significant (*p* < 0.05), and the nd indicates that it is below the minimum scale or resolution.

**Figure 5 plants-12-03091-f005:**
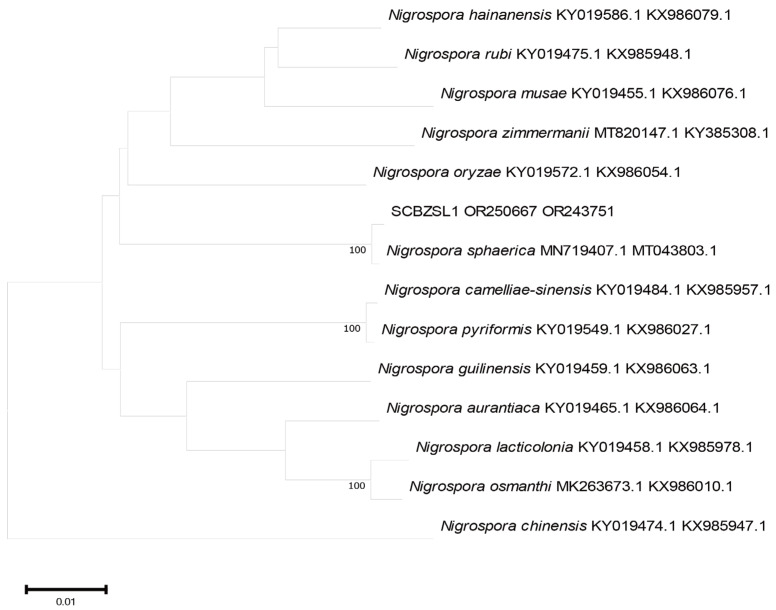
A phylogenetic tree based on ITS and BT joint construction.

**Figure 6 plants-12-03091-f006:**
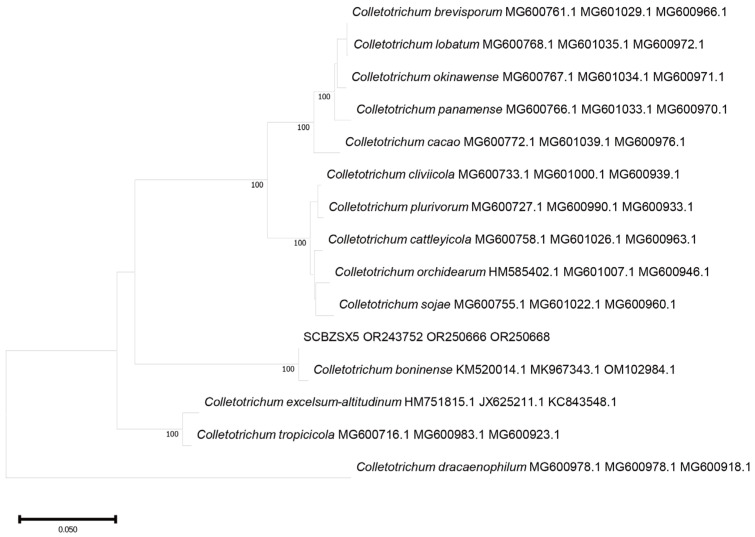
A phylogenetic tree based on ITS, BT, and ACT joint construction.

**Figure 7 plants-12-03091-f007:**
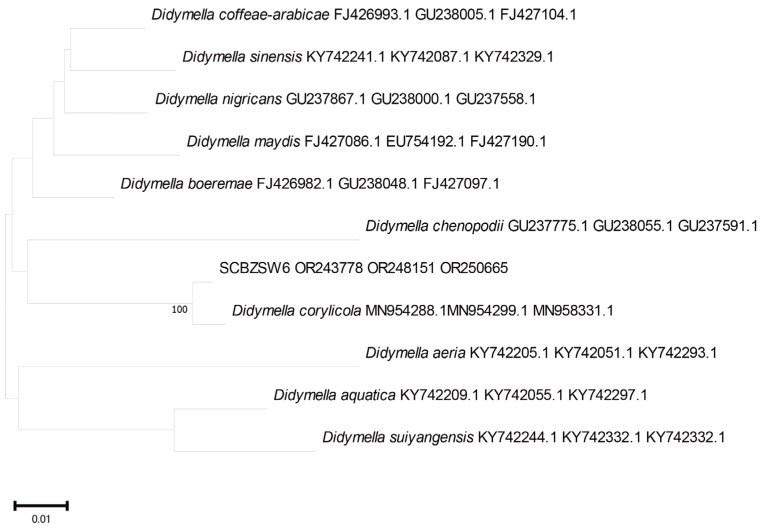
A phylogenetic tree based on ITS, LSU, and BT joint construction.

**Figure 8 plants-12-03091-f008:**
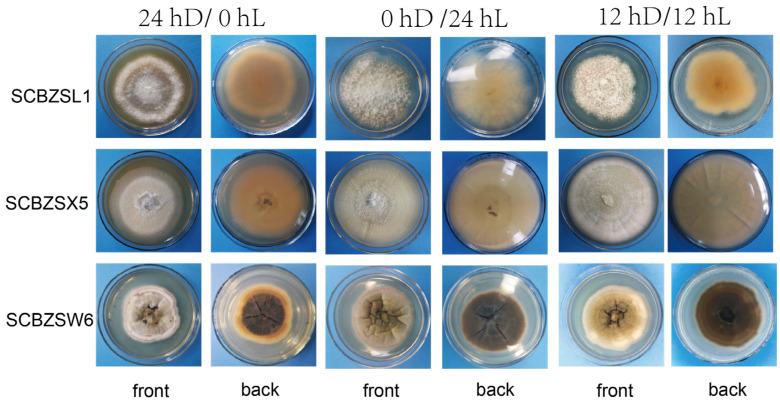
The first, second, and third lines represent the positive and negative growth morphology of SCBZL1, SCBZX5, and SCBZSW6 after 7 days of culture under three different light conditions.

**Figure 9 plants-12-03091-f009:**
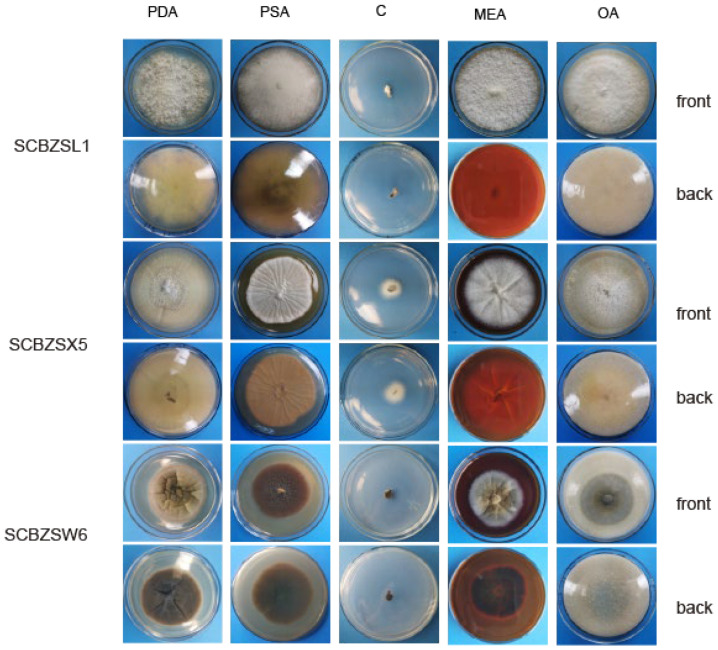
The first and second lines represent the positive and negative growth morphology of SCBZSL1 after 7 days of culture in different media, the third and fourth lines represent the positive and negative growth morphology of SCBZX5 after 7 days of culture in different media, and the fifth and sixth lines represent the positive and negative growth morphology of SCBZSW6 after 7 days of culture in different media.

**Figure 10 plants-12-03091-f010:**
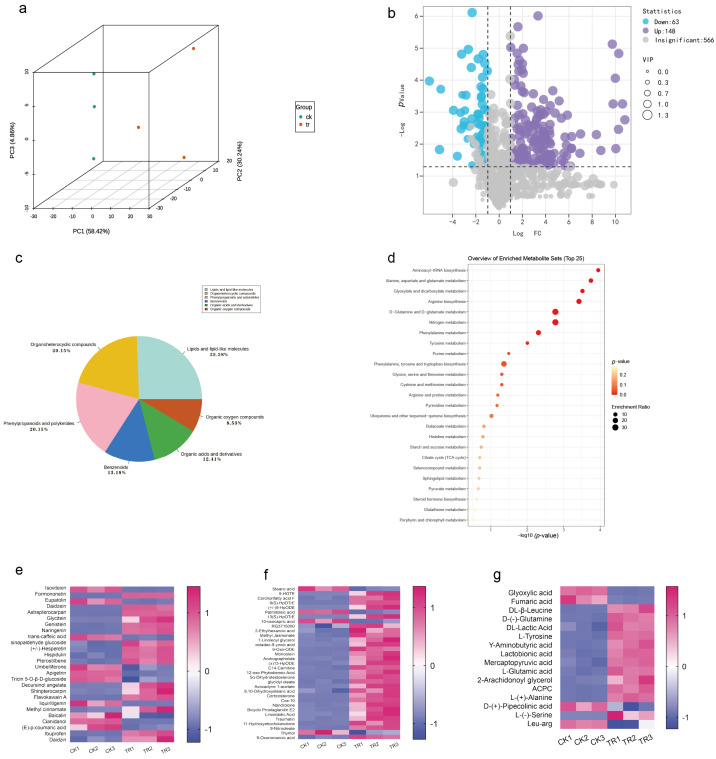
(**a**) Principal component analysis (PCA) of *sorghum–sudangrass* hybrids treated with SCBZSX5 and healthy *sorghum–sudangrass* hybrids. (**b**) Using GraphPad Prism 8.0 software to create volcano maps, the purple part indicates that the differential metabolites are up-regulated, the blue part indicates that the differential metabolites are down-regulated, and the gray part indicates the insignificant metabolites. (**c**) Differential metabolite classification chart created using Excel 2019. (**d**) Differential metabolism chemicals were analyzed using Metaboanalyst 5.0 for important metabolic pathways, the KEGG database was used to annotate and construct the pathway, the *x*-axis denotes the pathway impact, the *y*-axis represents pathway enrichment, the circle size denotes pathway to impact, and the circle color from red to yellow represents the *p*-value (*p* < 0.05) becoming smaller. (**e**–**g**) Using the online graphing tool Metware Cloud and GraphPad Prism 8.0 software, potential differential metabolites were screened according to different projected significance (VIP > 1 and *p* < 0.05), and differential metabolites’ heat maps of flavonoids and isoflavones, differential metabolites’ heat maps of lipid and lipid molecules, heat maps of differential metabolites of organic acids, and their derivatives were generated.

**Figure 11 plants-12-03091-f011:**
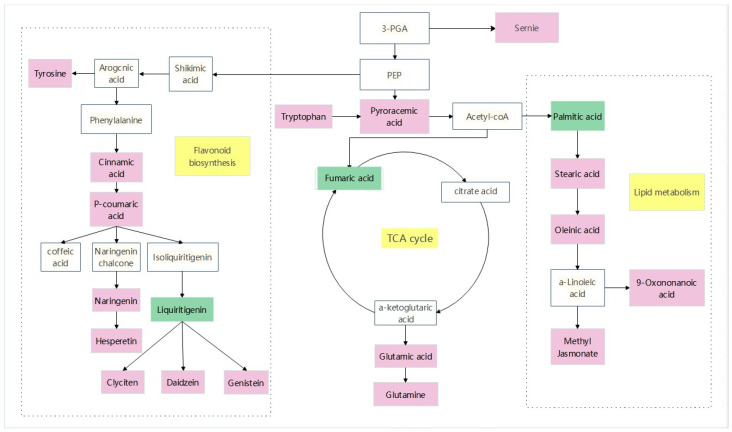
Analyze relevant data from MetaboAnalyst and KEGG websites to identify key differential metabolites, and then use Adobe Illustrator 2021 V25.3.1.390 mapping software to obtain pathway maps. The white box in the diagram represents an undetected metabolite. The pink box represents significant enrichment of metabolites. The green box represents a significant reduction in metabolites. The yellow box represents the metabolic pathway.

**Table 1 plants-12-03091-t001:** The number of rhizomes, root bulk density, plant height, and root length of *Sorghum–sudangrass* hybrids treated with different pathogenic fungi.

Group	Number of Rhizomes (Root)	Root Bulk Density (cm^3^)	Plant Height (cm)	Root Length (cm)
SCBZSL1	10.99 ± 0.56 ^b^	1.42 ± 0.58 ^b^	104.63 ± 0.34 ^a^	8.76 ± 0.75 ^a^
SCBZSX5	8.19 ± 0.88 ^c^	1.21 ± 0.49 ^b^	84.29 ± 0.66 ^b^	6.37 ± 0.46 ^b^
SCBZSW6	11.43 ± 0.64 ^a^	4.88 ± 0.54 ^a^	105.53 ± 0.49 ^a^	8.95 ± 0.38 ^a^
Control	12.89 ± 0.76 ^a^	4.97 ± 0.37 ^a^	113.57 ± 3.75 ^a^	9.33 ± 0.58 ^a^

Note: the same lowercase letters indicate no significant difference (*p* value > 0.05), while different lowercase letters indicate significant differences (*p* < 0.05).

**Table 2 plants-12-03091-t002:** The isolation strain, sequence registration number, plant host, and original location of the strain were used for phylogenetic analysis in this study.

	Species	Accession Numbers ^1,2^	Host	Locality	GenBank Accession Numbers ^3^			
					ITS	TUB2	ACT	LSU
SCBZSL1	*N. aurantiaca*	LC 7302	*Nelumbo* sp. (leaf)	China	KX986064	KY019465	-	KX986098
	*N. camelliae-sinensis*	LC 2710	*Castanopsis* sp.	China	KX985957	KY019484	-	-
	*N. chinensis*	LC 2696	*Lindera aggregata*	China	KX985947	KY019474	-	-
	*N. guilinensis*	LC 7301	*Nelumbo* sp. (stem)	China	KX986063	KY019459	-	-
	*N. hainanensis*	LC 6979	*Musa paradisiaca* (leaf)	China	KX986079	KY019586	-	-
	*N. lacticolonia*	LC 3324	*Camellia sinensis*	China	KX985978	KY019458	-	KX986105
	*N. musae*	CBS 319.34 *	*Musa paradisiaca* (leaf)	Australia	KX986076	KY019455	-	-
	*N. oryzae*	LC 6759	*Oryza sativa*	China	KX986054	KY019572	-	-
	*N. osmanthi*	LC 4350	*Osmanthus* sp.	China	KX986010	MK263673	-	-
	*N. pyriformis*	LC 4669	*Castanopsis* sp.	China	KX986027	KY019549	-	-
	*N. rubi*	LC 2698	*Rubus* sp.	China	KX985948	KY019475	-	KX986102
	*N. sphaerica*	-	*Tamarind* (leaf)	Malaysia	MT043803	MN719407	-	-
	*N. zimmermanii*	CBS 167.26	*-*	China	KY385308	MT820147	-	-
SCBZSX5	*C. brevisporum*	CBS 512.75	*Carica papaya*	Australia	MG600761	MG601029	MG600966	-
	*C. boninense*	CBS 123755	*Schefflera heptaphylla*	China	KM520014	MK967343	OM102984	-
	*C. cattleyicola*	CBS 170.49 *	*Cattleya* sp.	Belgium	MG600758	MG601026	MG600963	-
	*C. cliviicola*	CBS 125375 *	*Clivia miniata*	China	MG600733	MG601000	MG600939	-
	*C. cacao*	CBS 119297 *	*Theobroma cacao*	Costa Rica	MG600772	MG601039	MG600976	-
	*C. panamense*	CBS 125386 *	*Merremia umbellata*	Panama	MG600766	MG601033	MG600970	-
	*C. plurivorum*	CBS 125474	*Abelmoschus esculentus*	Japan	MG600727	MG600990	MG600933	-
	*C. excelsum—altitudinum*	CGMCC 3.15130 *	*Bletilla ochracea*	China	HM751815	JX625211	KC843548	-
	*C. lobatum*	IMI 79736 *	*Piper catalpaefolium*	Trinidad and Tobago	MG600768	MG601035	MG600972	-
	*C. orchidearum*	CBS 135131	*Cymbidium hookerianum*	China	HM585402	MG601007	MG600946	-
	*C. okinawense*	CBS 143246	*Carica papaya*	Japan	MG600767	MG601034	MG600971	-
	*C. sojae*	CBS H 21495	*Glycine max*	Iran	MG600755	MG601022	MG600960	-
	*C. dracaeneophilum*	CBS 119360	*Dracaena sanderana*	China	MG600711	MG600978	MG600918	-
	*C. tropicicola*	CBS 133174	*Citrus* sp.	Mexico	MG600716	MG600983	MG600923	-
SCBZSW6	*D. aeria*	CGMCC 3.18353	-	China	KY742205	KY742293	-	KY742051
	*D. aquatica*	CGMCC 3.18349	-	China	KY742209	KY742297	-	KY742055
	*D. boeremae*	CBS 109942	*Medicago littoralis*	Australia	FJ426982	FJ427097	-	GU238048
	*D. chenopodii*	CBS 128.93	-	The Netherlands	GU237775	GU237591	-	GU238055
	*D. coffeae-arabicae*	CBS 123380	*Coffea arabica*	Ethiopia	FJ426993	FJ427104	-	GU238005
	*D. corylicola*	CBS 146357	*Corylus avellana*	Italy	MN954288	MN958331	-	MN954299
	*D. maydis*	CBS 588.69	*Zea mays*	USA	FJ427086	FJ427190	-	EU754192
	*D. nigricans*	CBS 444.81	-	The Netherlands	GU237867	GU237558	-	GU238000
	*D. sinensis*	LC 8142	-	China	KY742241	KY742329	-	KY742087
	*D. suiyangensis*	LC 8144	-	China	KY742244	KY742332	-	KY742332
	*D. maydis*	CBS 588.69	*Zea mays*	USA	FJ427086	FJ427190	-	EU754192

^1^ CGMCC = China General Microbiological Culture Collection, Institute of Microbiology, Chinese Academy of Sciences, Beijing, China. CBS = Culture Collection of the Westerdijk Fungal Biodiversity Institute, Utrecht, the Netherlands. IMI = International Mycological Institute, CABI-Bioscience, Egham, Bakeham Lane, UK. LC = working collection of Lei Cai, housed at the Institute of Microbiology, Chinese Academy of Sciences, Beijing, China. ^2^ * = ex-type culture. ^3^ ITS = internal transcribed spacers and intervening 5.8S nrDNA. TUB2 = Beta-tubulin. ACT = partial actin gene. LSU = 28S nrRNA gene.

**Table 3 plants-12-03091-t003:** Effect of different items on spore production.

Item	Treatments	SCBZSL1	SCBZSX5	SCBZSW6
Different light regimes (h)	24 h/0	-	3.770 ± 0.028 ^a^	-
	0 h/24	-	4.735 ± 0.064 ^b^	-
	12 h/12	-	4.250 ± 0.004 ^c^	-
Different pH	5	-	5.156 ± 0.078 ^a^	-
	7	-	4.305 ± 0.132 ^bc^	-
	9	-	4.054 ± 0.062 ^c^	-
	11	-	-	-
Different temperature (°C)	5	-	4.074 ± 0.033 ^a^	-
	10	-	-	-
	15	-	4.537 ± 0.052 ^b^	-
	20	-	-	-
	25	-	4.785 ± 0.022 ^c^	-
	30	-	4.650 ± 0.070 ^bc^	-
	35	-	-	-
Different media	PDA	-	4.843 ± 0.064 ^a^	-
	C	-	-	-
	OA	-	-	-
	PSA	-	4.081 ± 0.023 ^b^	-
	MEA	-	-	-

Note: - indicates that there is no spore production under this condition. The data in the table indicate the value of the actual number of spores after lg conversion. The same lowercase letters indicate no significant difference (*p* > 0.05), while different lowercase letters indicate significant differences (*p* < 0.05).

**Table 4 plants-12-03091-t004:** Primers and sequences.

Gene	Primers	Sequence (5′-3′)	References
ITS	ITS1	TCCGTAGGTGAACCTGCGG	[43]
	ITS4	TCCTCCGCTTATTGATATGC	
Tub	Bt-2a	GGTAACCAAATCGGTGCTGCTTTC	[44]
	Bt-2b	ACCCTCAGTGTAGTGACCCTTGGC	
ACT	ACT512F	ATGTGCAAGGCCGGTTTCGC	[44]
	ACT783R	TACGAGTCCTTCTGGCCCAT	
LSU	LR0R	ACCCGCTGAACTTAAGC	[45]
	LR7	TACTACCACCAAGATCT	

## Data Availability

Not applicable.

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
