# Peer review of "First Report of Fungal Pathogens Causing Leaf Spot on Sorghum–Sudangrass Hybrids and Their Interactions with Plants"

_plants, 2023, doi:10.3390/plants12173091_

Round 1
Reviewer 1 Report
Well done
Author Response
Dear reviewer 1:
Thank you for your recognition of our work. We would like to express our sincere gratitude for reviewing our articles during your busy schedule, really appreciate your review. I wish you success in your work, peace and joy.
Kind regards,
Dr. Yuzhu Han
Reviewer 2 Report
The authors of this manuscript present interesting research on fungal pathogens causing leaf spot on sorghum sudangrass hybrids and their interactions with plants. Introduction and material and methods sections are well described. Results presented in tables and figures are clear and quite explicable. Due to the limited research as a first approach, I believe that the authors discuss and explain the findings of their work compering their work with other similar works in a well written discussion. The text needs very few revisions. Although Colletotrichum sp and other fungal pathogens causing leaf spot are well known for their effect on crop yield and have been studied in the past by many researchers, I think that this research study could add further interest to the researchers worlwide.
Title
Please check title format. I think that every first letter of each word must be capitalized.
Abstract
COMMENT:
The abstract describes well the purpose and the scope of this research work.
Introduction
Introduction section is well written and, in my opinion, give the appropriate information.
Line 32 Please use italic names in Sorhum bicolor and Sorghum sudanense. Please check and apply to the rest of the document.
Line 35 foliage [1]. Please leave gap…Check and apply to the rest of the document.
Line 57 characteristics. The
Results
Results are presented in tables and figures are clear and quite explicable.
Line 92 2.2. Morphological identification Please leave a gap between different lines. Please check and apply to the rest of the document
Table 1 Please apply italics to latin names of host
Material and Methods
Line 452 Table 3 (leave a gap) Please check and apply the rest of the manuscript
Discussion
Discussion section is well organized. The authors compare their findings with other similar research work. However, in my opinion discussion section could not contain any table and figures. Thus, I suggest figures 10 and 11 to be mentioned somewhere else in the manuscript.
Line 319 Sichuan Province,
Conclusions
References
COMMENT:
Please check reference list once again in order to be sure that is according to author’s instruction.

Author Response
Dear reviewer 2:
We are grateful for your valuable suggestions and recognition. Thank you again for spending a lot of valuable time patiently reviewing the manuscript. At the same time, thank you very much for affirming our research content, and we will consider adding researchers for further research in the future.We have solved the problem you raised one by one; all our changes of manuscript are marked up using the “Track Changes” function of MS Word. May happiness and health be with you always.
Kind regards,
Dr. Yuzhu Han
Responds to the reviewers' comments:
The authors of this manuscript present interesting research on fungal pathogens causing leaf spot on sorghum sudangrass hybrids and their interactions with plants. Introduction and material and methods sections are well described. Results presented in tables and figures are clear and quite explicable. Due to the limited research as a first approach, I believe that the authors discuss and explain the findings of their work compering their work with other similar works in a well written discussion. The text needs very few revisions. Although Colletotrichum sp and other fungal pathogens causing leaf spot are well known for their effect on crop yield and have been studied in the past by many researchers, I think that this research study could add further interest to the researchers worldwide.
Reply: Thank you for your recognition and support of our work, our article aims to study the pathogenesis, pathogenic mechanism and prevention methods of sorghum sudangrass hybrids leaf spot, we identify pathogenic bacteria through pathogenic return experiments, and find solutions through biological characteristics and metabolomic analysis, which are different from other research angles in plant diseases. We will conduct further research based on your suggestions to better solve the problem of the decline in yield of sorghum sudangrass hybrids caused by leaf spot.
Comments:
1、Please check title format. I think that every first letter of each word must be capitalized.
A: The first letters of all words in the title have been changed to uppercase.
2、Line 32 Please use italic names in Sorhum bicolor and Sorghum sudanense. Please check and apply to the rest of the document.
A: All the sorghum sudangrass hybrids in the article have been changed to italics. as required.
3、Line 35 foliage [1]. Please leave gap…Check and apply to the rest of the document.
A:The spaces between the reference number and the content of the article have been modified.
4、Line 57 characteristics. The.
A:The error in line 57 in the original manuscript has been corrected by line 58 in the revised version.
5、Line 92 2.2. Morphological identification Please leave a gap between different lines. Please check and apply to the rest of the document.
A: Followed the requirements in the article leave a gap between different lines and have checked the full text to modify in accordance with this requirement.
6、Table 1 Please apply italics to latin names of host.
A:Table 1 The Latin for host has been completely modified to italicize.
7、Line 452 Table 3 (leave a gap) Please check and apply the rest of the manuscript.
A: Gap have been left between tables and numbers throughout the text as required.
8、However, in my opinion discussion section could not contain any table and figures. Thus, I suggest figures 10 and 11 to be mentioned somewhere else in the manuscript.
A: The position of Pic 10 and Pic 11 has been modified to the Results section.
9、Line 319 Sichuan Province,
A:The error in line 319 in the original manuscript has been corrected by line 369 in the revised version.
10、Please check reference list once again in order to be sure that is according to author’s instruction.
A: It has been re-checked in the section on references.

Reviewer 3 Report
Colletotrichum boninense: Similar type of article has been published in J Fungi (Basel).( 2023 Jan; 9(1): 100.) where the authors have reported anthracnose of Sorghum caused by Colletotrichum sublineola. the authors should include the same species along with others to analyse the phylogentic relationship with other species and identification. Similar is my observation for other pathogens too. Didymella corylicola causes disease in Hazelnut. The repport is a new one for the sorghum-sudangrass hybrid so, needs thourough investigation like why they are changing theit host preference.
Author Response
Dear reviewer 3:
We are grateful for your valuable suggestions and recognition. Thank you again for spending a lot of valuable time patiently reviewing the manuscript. We have solved the problem you raised one by one; all our changes of manuscript are marked up using the “Track Changes” function of MS Word. May happiness and health be with you always.
Kind regards,
Dr. Yuzhu Han
Comments and Suggestions:
Colletotrichum boninense: Similar type of article has been published in J Fungi (Basel). (2023 Jan; 9(1): 100.) where the authors have reported anthracnose of Sorghum caused by Colletotrichum sublineola. the authors should include the same species along with others to analyse the phylogentic relationship with other species and identification. Similar is my observation for other pathogens too. Didymella corylicola causes disease in Hazelnut. The repport is a new one for the sorghum-sudangrass hybrid so, needs thourough investigation like why they are changing theit host preference.
A:First, we very much recognize the suggestions you have made to us. Regarding your suggestion that include the same species along with others to analyze the phylogenetic relationship with other species and identification. We modified the phylogenetic tree in the original manuscript to include Colletotrichum sublineola, Colletotrichum fructicola, which causes sorghum anthrax [1]、[2], Colletotrichum gloeosporioides, which causes yam anthrax [3], and Colletotrichum aenigma, which causes grape anthrax [4]. Also, regarding the further study of host preference issues you mentioned, we think this question is important. Because considering that although Colletotrichum boninense was first reported on sorghum sudangrass hybrids, we found that it is not a new fungus through phylogenetic tree comparison, and in many references we read earlier, it was found that Colletotrichum boninense can cause anthrax in a variety of plants, which is also mentioned in the discussion section of the article, so we currently speculated that it causes a wide range of diseases and there is no preference for the host. Of course, this is just our conjecture, your suggestions provide a good idea for our follow-up research, thank you again for your comments.
Reference:
- Koima, I.N.; Kilalo, D.C.; Orek, C.O.; Wagacha, J.M.; Nyaboga, E.N. Identification and Characterization of Colletotrichum Species Causing Sorghum Anthracnose in Kenya and Screening of Sorghum Germplasm for Resistance to Anthracnose. Journal of Fungi 2023, 9, 100, doi:10.3390/jof9010100.
- Zhao, W.; Hu, A.; Ren, M.; Wei, G.; Xu, H. First Report on Colletotrichum Fructicola Causing Anthracnose in Chinese Sorghum and Its Management Using Phytochemicals. Journal of Fungi 2023, 9, 279, doi:10.3390/jof9020279.
- Dentika, P.; Blazy, J.-M.; Alleyne, A.; Petro, D.; Eversley, A.; Penet, L. High Genetic Diversity and Structure of Colletotrichum Gloeosporioides s.l. in the Archipelago of Lesser Antilles. Journal of Fungi 2023, 9, 619, doi:10.3390/jof9060619.
- Yu, S.; Wang, J.; Chai, R.; Qiu, H.; Lu, Z.; Zhang, Z.; Li, L.; Wang, J.; Sun, G. Fluorescent Labeling of Peroxisome and Nuclear in Colletotrichum Aenigma. Journal of Fungi 2023, 9, 493, doi:10.3390/jof9040493.
